# Biodegradation of Oil by a Newly Isolated Strain *Acinetobacter junii* WCO-9 and Its Comparative Pan-Genome Analysis

**DOI:** 10.3390/microorganisms11020407

**Published:** 2023-02-06

**Authors:** Shijie Jiang, Qingfeng Fan, Zeying Zhang, Yunfeng Deng, Lihong Wang, Qilin Dai, Jin Wang, Min Lin, Jian Zhou, Zhijian Long, Guiqiang He, Zhengfu Zhou

**Affiliations:** 1School of Life Science and Engineering, Southwest University of Science and Technology, Mianyang 621010, China; 2Engineering Research Center of Biomass Materials, Ministry of Education, Southwest University of Science and Technology, Mianyang 621010, China; 3Key Laboratory of Agricultural Microbiome (MARA), Biotechnology Research Institute, Chinese Academy of Agricultural Sciences, Beijing 100081, China

**Keywords:** *Acinetobacter junii* WCO-9, lipase, oil degradation capabilities, pan-genome analysis, triglyceride degradation pathway

## Abstract

Waste oil pollution and the treatment of oily waste present a challenge, and the exploitation of microbial resources is a safe and efficient method to resolve these problems. Lipase-producing microorganisms can directly degrade waste oil and promote the degradation of oily waste and, therefore, have very significant research and application value. The isolation of efficient oil-degrading strains is of great practical significance in research into microbial remediation in oil-contaminated environments and for the enrichment of the microbial lipase resource library. In this study, *Acinetobacter junii* WCO-9, an efficient oil-degrading bacterium, was isolated from an oil-contaminated soil using olive oil as the sole carbon source, and its enzyme activity of ρ-nitrophenyl decanoate (ρ-NPD) decomposition was 3000 U/L. The WCO-9 strain could degrade a variety of edible oils, and its degradation capability was significantly better than that of the control strain, *A junii* ATCC 17908. Comparative pan-genome and lipid degradation pathway analyses indicated that *A. junii* isolated from the same environment shared a similar set of core genes and that the species accumulated more specific genes that facilitated resistance to environmental stresses under different environmental conditions. WCO-9 has accumulated a complete set of oil metabolism genes under a long-term oil-contamination environment, and the compact arrangement of abundant lipase and lipase chaperones has further strengthened the ability of the strain to survive in such environments. This is the main reason why WCO-9 is able to degrade oil significantly more effectively than ATCC 17908. In addition, WCO-9 possesses a specific lipase that is not found in homologous strains. In summary, *A. junii* WCO-9, with a complete triglyceride degradation pathway and the specific lipase gene, has great potential in environmental remediation and lipase for industry.

## 1. Introduction

Fumes, oily wastewater and oily solid waste generated by the catering industry have caused a series of environmental problems. Untreated oily wastewater enters drainage pipes easily, causing clogging, bad odors and other problems, which in turn, place a significant burden on urban sewage treatment, with solid oily waste further increasing treatment cost and degradation difficulty [1,2]. Traditional physicochemical methods (such as separation, landfill, flocculation, electrolysis, etc.) [3,4] for the treatment of waste oil are expensive and create secondary pollution problems [5]. One study showed that the degradation rate of hot pot waste oil by *Klebsiella pneumoniae* LZU10 was 37.7% in 48 h [6]. Ke et al. (2021) screened and constructed microbial flora for in situ biodegradation of waste cooking oil, and the degradation rate was 57.38% in 72 h, which was significantly higher than that of a single strain [7]. In recent years, research reports on the biodegradation treatment of urban waste cooking oil have continued to demonstrate the high efficiency and broad application prospects of bioremediation [8]. It is also worth mentioning that the use of microorganisms and bioenzymes to treat waste cooking oil and convert it into high-value resources such as bioenergy and fatty acids has shown great potential [9,10,11,12]. Kumari et al. (2017) obtained a yield of 6.6 mg/g fatty acid from waste oil pretreated with lipase and digested by *Penicillium chrysogenum* [13]. Modified *Rhizomucor miehei* lipase (RML) was used to catalyze the conversion of waste soybean oil into biodiesel, and this achieved a conversion rate of 81.7% within 24 h [14]. Biodegradation and bioconversion strategies for dealing with waste cooking oil are undoubtedly the green way to solve the kitchen waste problem.

*Acinetobacter* sp. is a group of bacteria that are naturally widely distributed [15], and they can be isolated from surface water [16], sewage pipes [17], soil, plants and animals [18]. With the advent of sequencing technology, a large number of members of *Acinetobacter* have been identified, and their systematic taxonomic status has been gradually developed [19]. Most of the *Acinetobacter* species in the natural environment are beneficial microflora and detergents of petroleum and heavy metal-contaminated systems [20,21]. A study found that *Acinetobacter* bacteria were more abundant in long-term oil-contaminated soil, accounting for 5.26%, and possessed a strong ability to biodegrade alkane mixtures (C9-C30) [22]. Cai screened a strain of *Acinetobacter* sp. from oil-contaminated sludge, and this degraded up to 79.94% of total petroleum hydrocarbons (TPH) in 10 days [23]. The *Acinetobacter* genus is currently best reported in terms of infection and resistance to coliphages and carbapenems, as represented by *A. baumannii* [24,25]. With the rapid development of the catering industry, the amount of waste cooking oil produced is unprecedentedly large, and the use of *Acinetobacter* has become a new choice for the microbial treatment of waste cooking oil. Gao et al. (2019) screened two strains of *Acinetobacter* from an oily environment for efficient oil degradation, and the degradation efficiency with regard to 2% (*v*/*v*) gutter oil exceeded 90% in 72 h [26]. To solve the problem of a low in situ degradation rate caused by the high oil content of food waste, Ke et al. (2021) isolated and screened a variety of oil-degrading strains, including *Acinetobacter,* from oily waste and constructed artificial microflora to improve the in situ degradation rate [27]. Lipase activity is an important reference for the screening of oil-degrading strains, and it is also an indispensable testing parameter for the biodegradation of waste cooking oil [8,27]. This indicates that lipase is an important participant in the biodegradation of kitchen waste and an important factor for the strains that are able to degrade waste oil. Cai et al. (2022) focused on changes in the lipase, protease and cellulase activities in the system during the biodegradation of kitchen waste, and the lipase activity was the highest in the degradation process as a whole, fully illustrating the importance of its role [28]. As an important participant in enzyme preparation, lipase plays a significant role in industrial production and environmental remediation [29,30]. For example, lipase is used to promote the degradation of waste oil by plant endophytic microorganisms [31] and lipase catalyzes the production of biodiesel from waste cooking oil [32]. Given the great potential of microbial degradation of kitchen waste oil and the rich and diverse resources of microbial strains, it is important to explore and utilize oil-degrading bacteria and bioenzymes in order to achieve efficient degradation and resource transformation of waste cooking oil and prevent environmental pollution [33].

Bacteria of *Acinetobacter* sp. have been continuously isolated from oily waste to show their potential application value, and there have been some studies concerning their application in the degradation of waste cooking oil and the mining of lipase from the *Acinetobacter* genus. However, there are few reports focusing on *Acinetobacter* oil-degrading bacteria with high-activity lipase, and the *Acinetobacter* oil degradation mechanism has not been studied at all. Based on high-throughput sequencing and data from online databases, it is possible to analyze the oil degradation pathways in *Acinetobacter* and explore the relevant genes. With the above objectives, our study screened and obtained from oil-contaminated soil the highly efficient oil-degrading bacterium, *A. junii* WCO-9, comparatively analyzed its oil degradation capabilities, determined the lipase activity of the strain, and revealed the specificity and availability of WCO-9 through pan-genome analysis, laying a foundation for the further transformation and utilization of WCO-9. This work provides a new impetus for the harmless treatment and resource utilization of kitchen waste.

## 2. Materials and Methods

### 2.1. Isolation and Screening of Strains

The oil-contaminated soil sample was collected from the Southwest University of Science and Technology, Mianyang, Sichuan Province (N 31.53791, E 104.69388). For the isolation and screening of lipase-producing strains, 5 g of the soil sample was weighed in 45 mL of physiological saline, incubated for 1 h at 30 °C while being shaken at 200 r/min, and then rested. A quantity of 1 mL of supernatant was added to 20 mL of enrichment medium (yeast paste 0.2 g/L, NaCl 0.5 g/L, Na_2_HPO_4_ 3.5 g/L, KH_2_PO_4_ 1.5 g/L, MgSO_4_-7H_2_O 0.5 g/L), and this was incubated for 24 h at 30 °C while being shaken at 200 r/min. After gradient dilution of the enrichment medium, 100 μL of 10^−6^~10^−8^ dilution was coated onto the oil assimilation plate containing rhodamine B (25 mL olive oil emulsion was mixed well with 175 mL LB medium, 200 μL 10% rhodamine B solution was added, and the mixture was shaken). After incubation at 30 °C for 2 d, a pure culture capable of producing degradation circles was obtained, named WCO-9. The single colony of strain WCO-9 was selected and inoculated in an LB liquid medium. Cells in the logarithmic growth phase were obtained by shaking at 200 r/min and incubating at 30 °C for 12 h and were then stored in glycerol tubes. Finally, the WCO-9 strain was identified and registered by the Guangdong Microbial Culture Collection Center with the serial number GDMCC No: 61851. The scanning electron microscopy observation, 16S rDNA analysis and whole genome sequencing were performed using cells from the logarithmic growth stage.

### 2.2. Phylogenetic Analysis of 16S rDNA and Housekeeping Genes

The total genomic DNA of the WCO-9 strain was extracted using a commercial bacterial whole genome extraction kit (Tiangen, Beijing, China). The 16S rDNA sequence of the strain was amplified using universal primer F27/R1492, and the purified 16S rDNA fragments were sequenced by BGI (Beijing, China). Sequencing of the housekeeping genes (*ileS*, *recA*, *rpoD*) was carried out by BioMarker (Beijing, China). The NCBI BLAST system (https://www.ncbi.nlm.nih.gov/, accessed on 26 June 2022) was used to compare and analyze the 16S rDNA and the housekeeping genes. MEGA 11.0 software was used to construct a phylogenetic tree using the neighbor-joining method, and bootstrap values were calculated based on 1000 replicates. Phylogenetic analysis identified *A. junii* ATCC 17908 (ACCC preservation number: 11037) as the closest relative to strain WCO-9, and this was purchased from ACCC as a control for comparative analysis.

### 2.3. Comparison of the Growth and Oil Degradation Capabilities of Strain WCO-9

LB liquid medium was used to explore the optimal growth, lipase production temperature and pH of the WCO-9 strain, and ρ-nitrophenyl decanoate (C10), ρ-nitrophenyl laurate (C12), ρ-nitrophenyl myristate (C14), ρ-nitrophenyl palmitate (C16) and ρ-nitrophenyl stearate (C18) was chosen as the substrate for the lipase activity assay [34]. Oil degradation assays were performed on rhodamine B oil plates [35] using corn oil, soybean oil, peanut oil, canola oil and olive oil mixed with a solution of 4% polyvinyl alcohol (PVA) in a 1:3 volume ratio and emulsified by sonication for 10 min. Then, 25 mL of the emulsion was added to 200 mL of medium and stained with 10% rhodamine B solution. The WCO-9 and *A. junii* ATCC 17908 strains were activated by plate streaking, and single colonies were incubated in a liquid medium at 30 °C and shaken at 200 r/min for 12 h to prepare a seed solution, which was then inoculated in fresh liquid medium starting with an OD_600_ of 0.1. When both strains reached an OD_600_ of 0.7 ± 0.05, the bacterial concentrations of the two strains were adjusted to the same OD_600_ value. The oil plate was punched, and 10 μL of the two bacterial solutions with the same OD_600_ value was injected into the well and incubated at 30 °C for 4 days. The plate was observed under UV light, and the size of the transparent circle was measured as the parameter of oil degradation capabilities. The oil degradation activity was tested by GB/T_23535-2009 method with the five oils mentioned above.

### 2.4. Whole Genome Sequencing and Pan-Genome Analysis

The extraction and sequencing of the bacterial WCO-9 genome were performed by BioMarker in Beijing based on the Nanopore sequencing platform. The generated dataset corresponded to a sequencing depth of 100× The assembly of genomic data and correction of assembly results were performed using Canu v1.5 and Racon v3.4.3; gene prediction was performed using Prodigal v2.6.3; rRNA and tRNA were predicted using Infernal v1.1.3 and tRNAscan-SE v2.0 from non-redundant protein (NR); and Swiss-Prot, Pfam, Clusters of Orthologous Group (COG), Gene Ontology and Kyoto Encyclopedia of Genes and Genomes (KEGG) databases were used to obtain functional annotation of WCO-9 genes.

To further clarify the taxonomic status of strain WCO-9 and understand its specificity, pan-genomic analysis was performed using the genomes of 11 different *A. junii*. Using a bacterial pan-genome analysis tool (BPGA) channel with a homology sequence threshold of 50%, the core genome of *A. junii* was calculated based on the established functional model, and a neighbor-joining phylogenetic tree between the strains was derived based on core homologous proteins. The average nucleotide identity between *A. junii* strains was calculated using the BLAST (ANIb) and MUMmer (ANIm) algorithms, and the DNA–DNA hybridization (DDH) results of WCO-9 with the other 10 strains were obtained via numerical simulation using the Genome-to-Genome Distance Calculator (GGDC 2.1) (http://ggdc.dsmz.de/ (accessed on 26 June 2022)).

### 2.5. Analysis of Lipase Genes and Triacylglycerol Degradation Pathway

To discover the high lipase activity and lipid degradation capability of WCO-9, several lipase, lipase chaperone and triacylglycerol metabolic pathway genes of WCO-9 were analyzed. The lipase and lipase chaperone genes of strain WCO-9 were obtained by BLAST comparison with databases such as NR, Swiss-Prot, COG, Pfam, etc. The triacylglycerol degradation pathway of WCO-9 was obtained by a BLAST comparison of the sequenced related protein sequences with the sequences included in the KEGG database to find the most similar sequences. The sequence annotation information, the corresponding KO number, and the position in the metabolic pathway corresponding to the KO also represent the annotation information for the corresponding genes in the sequenced genome.

According to the phylogenetic tree of the core genes, *A. junii* ATCC 17908, *A. junii* CAM121, *A. junii* lzhX15, *A. junii* NIPH182 and *A. junii* YR7 belonged to different branches of the evolutionary tree; therefore, these strains were selected for the whole genome comparison with strain WCO-9. Genome-wide comparisons were carried out using the BLAST Ring Image Generator (BRIG) and image generation was performed using CGView with gene BLAST parameter values of 1 × 10^−5^ and gene identity thresholds of 70% (upper identity) and 50% (lower identity). Information in the triacylglycerol degradation pathway relating to lipase genes and other key enzyme genes of the WCO-9 strain were annotated in the outermost circle.

### 2.6. Statistical Analysis

The results were analyzed with GraphPad Prism 8.0.2, Origin2021 and SPSS Statistics 25 soft-mare. Differences in means between groups were compared for statistical significance at *p* < 0.05.

## 3. Results

### 3.1. Screening and Identification of Strain WCO-9

The strain WCO-9 was isolated from an oil-contaminated soil, and the isolate produced a distinct degradation circle on agar plates containing olive oil supplemented with rhodamine B (Figure 1A). Cultivation of this strain was improved in LB liquid medium at 30 °C, shaken at 200 r/min. Colonies of this strain were circular, convex, moist and opaque on the agar plates after 1 day of incubation at 30 °C. Staining experiments revealed that the isolate was Gram-negative (Appendix A). The cell morphology was spherical or in the form of short rods, 0.6–1.0 μm long and 0.4–0.7 μm wide without flagella (Figure 1B). The complete 16S rDNA sequence (1537 bp) of WCO-9 was used in the NCBI nucleotide database for comparative analysis and construction of the phylogenetic tree. The results showed that WCO-9 is an *A*. *junii* with 100% similarity to *A. junii* ATCC 17908 and less than 96% similarity to other genera of bacteria. The 16S rDNA phylogenetic analysis and partly housekeeping genes (*ileS*, *recA*, and *rpoD*) revealed that strain WCO-9 was clearly differentiated from other members of the *A. junii* family (Figure 2, Appendix A). The 16S rDNA phylogenetic analysis showed that WCO-9 clustered with *A. junii* and formed a monophyletic branch, which was related to *A. junii* ATCC 17908 (Figure 2).

### 3.2. Growth Characteristics and Oil Degradation Capabilities of Strain WCO-9

The growth and enzyme production of the strains in LB medium were investigated under different temperature and pH conditions. In essence, the increase in biomass of the strains was positively correlated with the increase in lipase activity. In LB medium with a pH of 7.0, strain WCO-9 was able to grow at 20, 25, 30, 35 and 40 °C, and 30 °C was the best temperature for growth and enzyme production temperature. When the temperature exceeded 30 °C, the growth and enzyme production ability of WCO-9 decreased sharply (Figure 3A). The WCO-9 strain was able to grow in a pH range of 5.0–10.0 and could not grow at pH 4.0. Its optimal growth and enzyme production was at pH 6.0 (Figure 3B). The lipase activity of WCO-9 was excellent, and the hydrolytic activity of ρ-NPD substrate produced by WCO-9 was 3040 U/L under optimal growth and enzyme production conditions.

The ability of WCO-9 and ATCC 17908 to degrade various oils was analyzed using agar plates containing olive oil supplemented with rhodamine B, and the results showed that strain WCO-9 had significant degradation effects on all five oils and was significantly better than strain ATCC 17908 (Table 1, Appendix A).

The degradation ability of strain WCO-9 was most evident for peanut oil, whereas its ability to degrade olive oil was relatively weak. Among the five oils, the lowest degradation activity for olive oil was 2833 U/L, and that for other oils was above 3000 U/L. The lipase activity of the WCO-9 strain was five times higher than that of the control strain. The results indicated that strain WCO-9 has potential application in the degradation of waste cooking oil.

### 3.3. Genome Comparison and Pan-Genome Analysis of WCO-9

*A. junii*, as a major member of the genus *Acinetobacter*, is widely distributed in the natural world. Genomic data relating to 81 strains of *A. junii* have been published in the NCBI database, of which 10 were selected based on the comparison results of 16S rDNA sequences and complete genome sequencing of WCO-9. The 10 strains were selected in combination with other sources and genomic information in the database. Eleven strains of *A. junii* were isolated, mainly from the soil, lake, sewage and the human environment, with genome size ranging between 3.19 and 3.78 Mb and GC content between 38.6% and 39%. Of these, strain WCO-9 had the smallest genome size and number of genes (Table 2).

We calculated the ANIb and ANIm of the 11 *A. junii* genomes and the DNA hybridization values of WCO-9 with the genomes of the other 10 strains (Table 3). The ANIb value of strain WCO-9 with *A. junii* lzh-X15 was 98.18% (98.16%), and the ANIm values with the rest of the strains were above 97%, which is higher than the species cut-off threshold of 95–96%. The DDH values of strain WCO-9 with the remaining 10 strains were all above the DDH threshold for the delineation of new species (70%).

A comparative analysis was performed between strain WCO-9 and 10 strains of *A. junii*, setting a sequence similarity threshold of 50% and calculating the core genes of the *A. junii* genome (Figure 4A). The genome of the 11 strains contained 35,215 genes, 2229 of which were core genes. The WCO-9 strain possessed 102 specific genes, accounting for 3.4% of the number of genes. KEGG annotation revealed that the specific genes were concentrated in metabolism and environmental information processing, primarily amino acid metabolism, carbohydrate metabolism, signal transduction, membrane transport, folding, sorting and degradation (Appendix A). COG annotation showed a higher proportion of specific genes in transcription, replication, recombination and repair, and general function. Phylogenetic analysis of the core genes showed that the 11 strains of *A. junii* were divided into four sub-evolutionary branches (Figure 4B), in which WCO-9 and *A. junii* lzh-X15 were both isolated from soil samples and clustered together in the evolutionary tree, whereas *A. junii* ATCC 17908 and *A. junii* CIP 107470, which were both isolated from liquid environments, were also grouped together and divided into different sub-branches because of the different compositions of the liquid environments.

### 3.4. Comparison of Lipase Genes and Triacylglycerol Metabolic Pathways

In order to further clarify the efficient lipid degradation mechanism of WCO-9, we used genome annotation to obtain all the lipases and chaperones in the WCO-9 genome. This showed that 11 lipase genes and 3 lipase chaperone genes were dispersed on the genome. The lipase chaperone *lif1700* and lipases *lip1701* and *lip1702,* and the lipase chaperones *lif2659* and lipase *lip2660* were separately arranged without spacers to form a gene cluster (Figure 5A). The degradation of triacylglycerol was initiated by triacylglycerol lipase (TGL). In one of the degradation pathways, triacylglycerol was hydrolyzed by TGL to 1,2-diacylglycerol after the third ester bond and then transformed into 1,2-diacylglycerol-3-phosphate by diacylglycerol kinase (*dgkA*) to enter the oxidative metabolism of phosphoglycerate. In another pathway, triacylglycerol was degraded by TGL to monoacylglycerol after the third and first ester bonds were hydrolyzed. The other degradation pathway was the stepwise degradation of triacylglycerol into monoacylglycerol by TGL with ester bonds at positions 3 and 1, further degradation by monoacylglycerol lipase (MGLL), and then from glycerol into 1,2-diacylglycerol-3-phosphate after three steps of activation into a phosphoglycerol metabolism (Figure 5B).

Five strains from each branch of the core evolutionary tree were selected for comparison with WCO-9 for lipase, chaperone and triacylglycerol degradation pathway genes. The results showed that 10 lipase and 3 lipase chaperone genes could be found in other strains of the *A. junii* family; ATCC 17908 lacked one lipase and one lipase chaperone compared with WCO-9; the sequence identity of two lipases and one lipase chaperone was less than 50% compared with WCO-9, and the two key initiator genes of the triacylglycerol degradation pathway (TGL2) were missing in all strains with the exception of strain NIPH 182 (Figure 6). Strain WCO-9 had one specific lipase gene (*lip1931*). Lip1931 was located on a gene island in the genome of strain WCO-9, and one gene each to the left (*GE_1930*) and right (*GE_1932*) of *lip1931* had no corresponding gene in the homologous strain. The rest of the genes on the gene island could be found in the comparison strains (Figure 7).

## 4. Discussion

*Acinetobacter* sp. strains live in diverse environments and, when isolated from their specific environments, have a significant lipid degradation capability and are, therefore, valuable in environmental bioremediation [36,37]. Our study used SEM to examine the cell morphology of *A. junii*, which revealed two morphologies, short rod-shaped and spherical, with interconnected cells, consistent with the results of the study carried out by Nandi et al. (2020) [38]. *A. junii*, as a branch of the large family of the *Actinobacter* genus, has been reported primarily as a pathogenic bacterium [39,40]. In an early study of *A. junii* for environmental remediation, Hrenovic et al. (2011) immobilized the *A. junii* DSM 1532 strain on natural zeolitic tuff to remove phosphorus from municipal wastewater [41]. Since then, there has been a gradual increase in the number of studies concerning the use of *A. junii* for environmental waste removal, including in the treatment of nitrogenous wastewater [42] and the degradation of phenol [43]. Recent studies also confirmed that bioremediation techniques using *A. junii* in combination with plants could improve waste degradation rates [44,45].

In this study, we found that the lipase genes of *A. junii* are abundant in quantity and variety, but there are few reports concerning the use of *A. junii* for oil degradation. The strains WCO-9 and ATCC 17908 have the closest 16S rDNA sequence identity, but they display significant differences in their oil degradation abilities (Table 1, Appendix A). A comparison of lipase and lipase chaperone genes revealed that the ATCC 17908 strain lacked one secretory lipase and one secretory lipase chaperone compared with WCO-9. In addition, the sequence consistency of two lipases (TGL2) and two lipase chaperones of ATCC 17908 was less than 70% compared with WCO-9, resulting in insufficient extracellular lipase activity in the ATCC 17908 strain. We compared the genes in the WCO-9 triacylglycerol degradation pathway with five other strains, including ATCC 17908. The results showed that four strains, including ATCC 17908, lacked the TGL2 lipase, which is the initiator for triacylglycerol degradation, and could not directly utilize triacylglycerol; thus, ATCC 17908 was not effective in lipid degradation. We speculated that most of the strains in the *A. junii* family were not capable of oil degradation, which is the main reason why there are few studies on lipid degradation in *A. junii*, and this suggests that WCO-9 is a special strain in this family by virtue of its oil degradation capabilities.

Lipase chaperones play an important role in the formation of lipase structures and the improvement of lipase activity. The lipase chaperones in the WCO-9 strain are closely aligned with multiple lipases, suggesting that such an arrangement is conducive to the more efficient formation of the advanced lipase structure of strain WCO-9, which exhibits high lipase activity. The chaperones also assist in the folding and secretion of lipase, further improving the oil degradation capability. Studies by Zheng et al. (2012) and others support this view [46,47]. Pan-genome analysis revealed that the strains from the same types of environment share a similar set of core genes and are clustered in the same evolutionary branch. The species has an extremely strong environmental response mechanism to cope with different environmental stresses, which explains the wide distribution of *A. junii* in different natural environments. The strain genome has a high proportion of genes specific to environmental information processing and metabolic functions, and the genome has a high degree of environmental plasticity. The presence of waste oil increased the expression of the lipase gene in strain WCO-9, which demonstrated higher lipase activity than the homologous strain ATCC 17908. To further improve the degradation and utilization of oils, the bacterial genome underwent recombination, mutation or gene exchange between different bacterial species. Therefore, we suggest that the significant oil degradation capabilities of WCO-9 are attributable to the long-term environmental stress caused by oil pollution and are a result of the adaptive evolution of bacteria under environmental stress. The same conclusion has been reached by the authors of previous studies [48,49,50]. The widespread use of lipases in a variety of fields has led to the expansion of their market, and the demand for more active and environmentally friendly lipases, and specific lipases, has been increasing [51]. *Acinetobacter* sp. has been intensively studied as an important source of alkaline and low-temperature lipases [52,53,54]. During its long-term evolution, strain WCO-9 produced a specific lipase gene, which belongs to a novel lipase, the biological function and physicochemical properties of which have never been studied and have a high mining value.

## 5. Conclusions

In this study, we isolated a bacterial strain with significant oil degradation capabilities from an environment contaminated with waste cooking oils. We identified it as *A. junii* using 16S rDNA and genome sequencing and named it *A. junii* WCO-9. We investigated the growth conditions for strain WCO-9 and found that its growth characteristics and lipase activity were optimal at 30 °C and pH 6.0. The hydrolysis activity of the lipase produced by the WCO-9 to the ρ-NPD substrate was about 3000 U/L. In order to investigate whether *A. junii* members were generally capable of lipid degradation, *A. junii* ATCC 17908 and WCO-9 were used in degradation tests on four edible vegetable oils, and it was found that WCO-9 had significant degradation capabilities for all four oils, whereas ATCC 17908 had almost no degradation effects. In summary, the long-term oil pollution environment was the external driving force for the formation of high oil degradation capacity of strain WCO-9. Under the pressure of oil, a large number of favorable mutations occurred in the genome of WCO-9, and a complete triglyceride degradation pathway was formed. The lipase gene structure of WCO-9 was different from that of the same species strain, so the oil degradation performance of WCO-9 was significantly better than that of the control strain. Lip1931, a unique secreted lipase of WCO-9, was inserted into the genome of WCO-9, and in addition, the close arrangement of the lipase chaperone and lipase on the genome differed from that of the same species strain, resulting in significantly higher lipase activity than that of the control strain. The degradation ability and high lipase activity of WCO-9 indicate its research value in environmental remediation and lipase gene mining. In addition, a total of 10 lipases of 4 types were found in the other strains of the family, which are rich in lipase genes and are an important source of microbial lipases. This study revealed the genetic factors for the efficient oil degradation capabilities of the WCO-9 strain, further confirmed the environment-driven evolution of *A. junii* and demonstrated the utilization value of strain WCO-9 in the preparation of mixed bacterial agents for waste oil degradation and environmental bioremediation. Furthermore, it laid a foundation for microbial lipase gene mining and strain modification based on the triacylglycerol pathway.

## Figures and Tables

**Figure 1 microorganisms-11-00407-f001:**
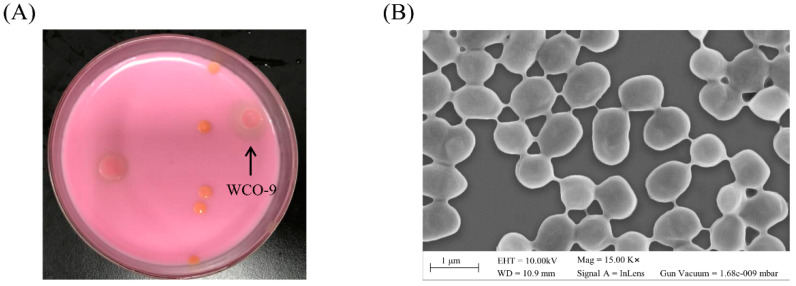
Colony and cell morphology of strain WCO-9: (**A**) colony morphology of strain WCO-9 grown on the oil assimilation plate containing rhodamine B for 24 h at 30 °C; (**B**) scanning electron microscopy (SEM) showing the morphology of strain WCO-9. Bar: 1 μm.

**Figure 2 microorganisms-11-00407-f002:**
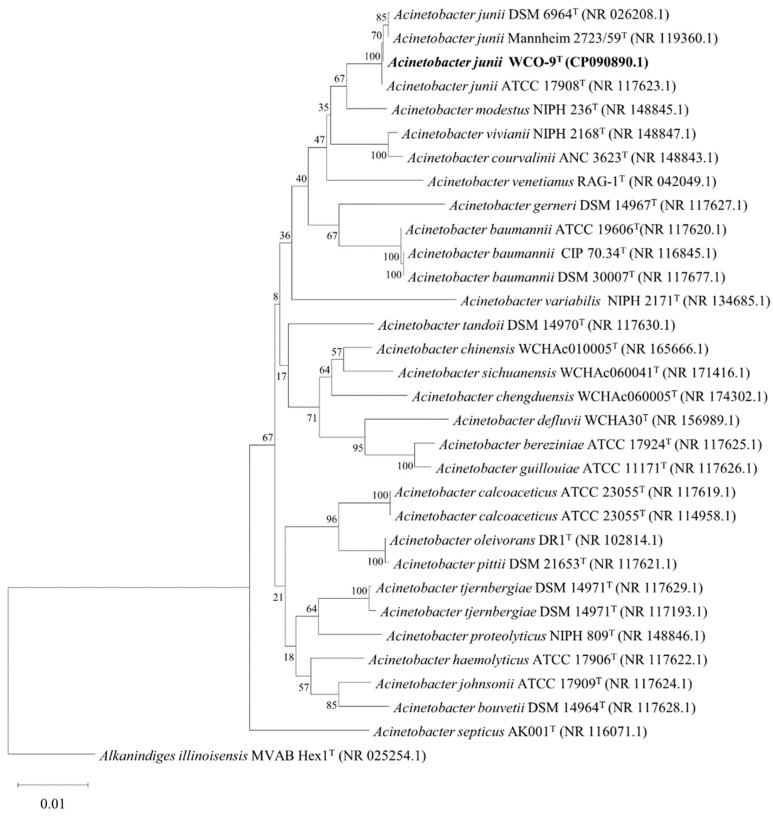
Phylogenetic tree based on 16S rDNA sequences reconstructed using the neighbor-joining method and using the outgroup strain *Alkanindiges illinoisensis* MVAB Hex1 as a tree root. This tree shows the phylogenetic relationship between strain WCO-9 and closely related species. Bootstrap percentages are based on 1000 replications. Bar: 0.01 substitutions per nucleotide position.

**Figure 3 microorganisms-11-00407-f003:**
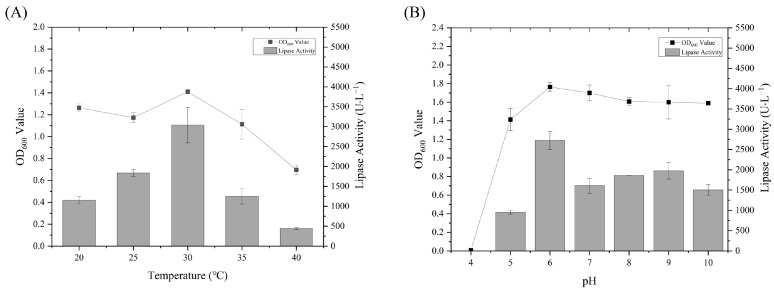
Growth characteristics and lipase activity of strain WCO-9: (**A**) growth of WCO-9 and lipase activity at different temperatures; (**B**) growth of WCO-9 and lipase activity under different pH conditions.

**Figure 4 microorganisms-11-00407-f004:**
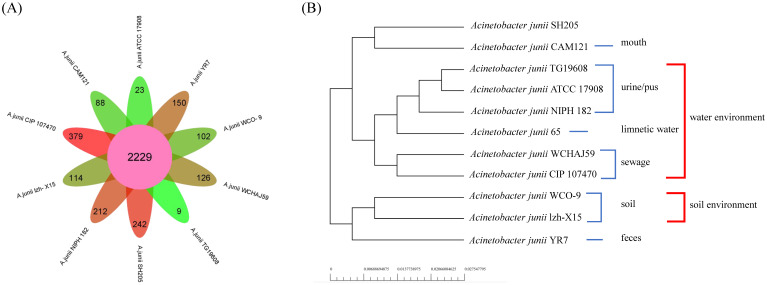
The pan-genome analysis and genome phylogenetic tree of strains belonging to *A. junii*: (**A**) petal diagram of the pan-genome showing the number of core genes of the species in the center and the number of petals corresponding to the specific genes of each *A. junii* strain; (**B**) phylogenetic tree constructed using the neighbor-joining method based on 2229 core protein genes.

**Figure 5 microorganisms-11-00407-f005:**
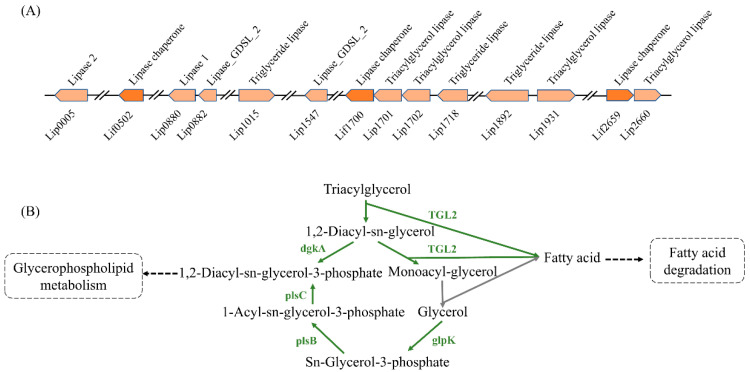
Lipase, chaperone genes and triacylglycerol degradation pathways of WCO-9: (**A**) distribution of lipase and chaperone genes in the genome of strain WCO-9; (**B**) triacylglycerol degradation pathway of strain WCO-9, the green arrows indicate the key metabolic pathway enzymes present in strain WCO-9.

**Figure 6 microorganisms-11-00407-f006:**
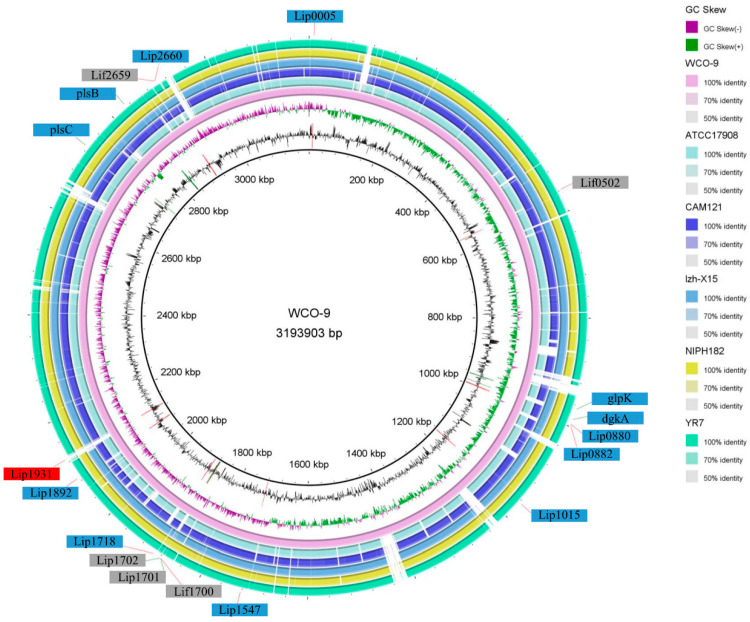
Comparison of lipase, chaperone and triacylglycerol degradation pathway genes between five strains of WCO-9 and other *A. junii*. The outermost circle is the annotation of lipase, lipase chaperone and triacylglycerol degradation pathway genes. The blue box indicates that the gene is present in all 11 strains, and the grey box indicates consistency of this gene in the ATCC 17908 genome below 50% or not present compared with the WCO-9 strain, and the red box shows the specific lipase gene for strain WCO-9.

**Figure 7 microorganisms-11-00407-f007:**
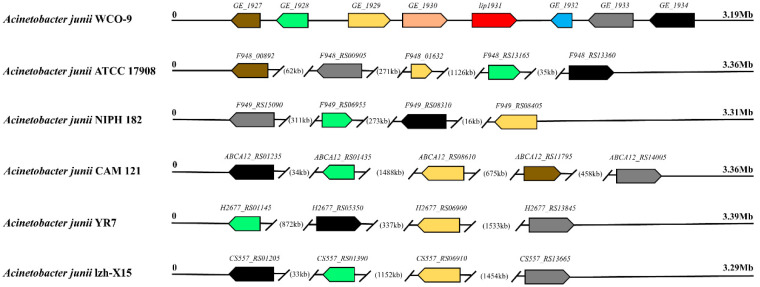
Comparison of the two flanking genes of the WCO-9 strain-specific lipase gene (*lip1931*). The same color represents the same gene.

**Table 1 microorganisms-11-00407-t001:** Comparison of lipase activity of WCO-9 and ATCC 17908.

Substrate	Activity of WCO-9	Activity of ATCC 17908	WCO-9’s Diameter of Degradation Circle	ATCC 17908’s Diameter of Degradation Circle
**Natural oils**	**(U/L)**	**(U/L)**	**(mm)**	**(mm)**
Corn oil	3358 ± 156	ND	23 ± 0.5	ND
Peanut oil	4219 ± 183	ND	25 ± 0.5	ND
Canola oil	3013 ± 134	ND	22 ± 0.5	ND
Soybean oil	3145 ± 173	ND	21 ± 0.5	ND
Olive oil	2833 ± 166	ND	21 ± 0.5	ND
**ρ-NP ester**	**(U/L)**	**(U/L)**		
ρ-nitrophenyl decanoate (C10)	2435 ± 177	ND	−	−
ρ-nitrophenyl laurate (C12)	2012 ± 90	177 ± 20	−	−
ρ-nitrophenyl myristate (C14)	1388 ± 50	119 ± 9	−	−
ρ-nitrophenyl palmitate (C16)	288 ± 14	46 ± 7	−	−
ρ-nitrophenyl stearate (C18)	161 ± 7	25 ± 13	−	−

Note: “ND” indicates that no oil degradation was observed. “−” indicates that the results had not been tested or could not be tested.

**Table 2 microorganisms-11-00407-t002:** General features of *A. junii* genomes used in this study.

Strain	Assembly No.	Level	Size (Mb)	GC (%)	No. of Genes	No. of Proteins	Isolation Source
*A. junii* WCO-9	CP090890.1	complete	3.19	38.6	3012	2890	oil-contaminated soil
*A. junii* ATCC 17908	APPX00000000.1	scaffold	3.36	38.9	3185	2947	urine
*A. junii* CAM121	CP068253.1	complete	3.36	38.8	3224	2898	mouth
*A. junii* 65	CP019041.1	complete	3.38	38.6	3211	2956	limnetic water
*A. junii* CIP 107470	APPS00000000.1	scaffold	3.78	38.6	3601	3302	activated sludge plant
*A. junii* lzh-X15	CP024632.1	complete	3.29	38.8	3069	2954	soil
*A. junii* NIPH 182	APPW00000000.1	scaffold	3.31	38.5	3174	3047	phlegmon pus
*A. junii* SH205	ACPM00000000.1	scaffold	3.46	39.0	3279	3079	missing
*A. junii* TG19608	AMJF00000000.1	contig	3.26	38.7	3108	2921	urine
*A. junii* WCHAJ59	CP028800.2	complete	3.35	38.9	3202	2968	sewage
*A. junii* YR7	CP059558.1	complete	3.39	38.6	3150	3023	feces

**Table 3 microorganisms-11-00407-t003:** ANIb, ANIm and DDH values of strain WCO-9 and 10 strains of the same species.

	Strain	1	2	3	4	5	6	7	8	9	10	11
ANIb												
1	*A. junii* WCO-9	*	97.18	97.23	97.23	98.18	97.97	97.09	94.38	97.06	97.18	97.95
2	*A. junii* ATCC17908	97.03	*	97.39	97.13	96.99	97.16	97.57	97.11	97.76	98.21	97.17
3	*A. junii* CAM121	97.12	97.30	*	97.00	97.07	97.14	97.04	94.35	97.63	97.56	97.05
4	*A. junii* CIP107470	96.92	96.76	96.88	*	96.82	97.13	96.80	93.89	96.95	96.96	96.92
5	*A. junii* lzh-X15	98.16	97.06	97.20	97.17	*	98.13	97.15	94.80	97.11	97.21	98.01
6	*A. junii* NIPH182	97.95	97.17	97.18	97.26	98.10	*	97.05	94.70	97.11	97.14	97.89
7	*A. junii* SH205	97.01	97.46	97.07	96.95	97.08	96.98	*	94.12	97.55	97.43	96.96
8	*A. junii* TG19608	97.00	100	96.86	97.38	96.52	97.51	96.55	*	96.63	97.45	96.95
9	*A. junii* WCHAJ59	97.05	97.82	97.70	97.17	97.07	97.17	97.51	93.44	*	97.84	97.09
10	*A. junii* 65	97.08	98.17	97.67	97.08	97.12	97.12	97.38	94.63	97.80	*	97.11
11	*A. junii* YR7	97.94	97.19	97.15	97.10	98.01	97.93	96.94	94.90	97.05	97.16	*
ANIm												
1	*A. junii* WCO-9	*	97.84	97.87	97.88	98.37	98.24	97.93	97.50	97.93	97.82	98.24
2	*A. junii* ATCC 17908	97.85	*	98.12	98.04	97.84	97.82	98.27	100	98.35	98.51	97.87
3	*A. junii* CAM121	97.87	98.13	*	97.94	97.95	97.92	98.12	97.40	98.42	98.17	97.92
4	*A. junii* CIP 107470	97.88	98.04	97.94	*	97.91	97.91	98.16	97.59	98.06	98.02	97.85
5	*A. junii* lzh-X15	98.37	97.83	97.95	97.92	*	98.32	97.93	97.61	97.93	97.89	98.29
6	*A. junii* NIPH182	98.24	97.82	97.92	97.91	98.32	*	97.89	97.69	97.91	97.86	98.30
7	*A. junii* SH205	97.93	98.26	98.12	98.16	97.93	97.89	*	98.39	98.28	98.31	97.91
8	*A. junii* TG19608	97.50	100	97.40	97.59	97.61	97.69	98.38	*	98.32	98.50	97.78
9	*A. junii* WCHAJ59	97.93	98.36	98.42	98.06	97.93	97.91	98.28	98.32	*	98.40	97.90
10	*A. junii* 65	97.82	98.51	98.17	98.02	97.89	97.86	98.32	98.50	98.40	*	97.87
11	*A. junii* YR7	98.24	97.87	97.92	97.85	98.29	98.30	97.91	97.75	97.90	97.87	*
DDH												
1	*A. junii* WCO-9	*	78.8	78	78.8	85.8	83.8	79.4	77.9	79.1	78.5	84.2

Note: “*” indicates that it cannot be compared with itself.

## Data Availability

The datasets for this research can be found in online repositories. The NCBI GenBank accession numbers for the complete genome of strain WCO-9 are CP090890.1 (https://www.ncbi.nlm.nih.gov/nuccore/CP090890.1, accessed on 17 January 2022).

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
