# Peer review of "Biodegradation of Oil by a Newly Isolated Strain Acinetobacter junii WCO-9 and Its Comparative Pan-Genome Analysis"

_microorganisms, 2023, doi:10.3390/microorganisms11020407_

Round 1

Reviewer 1 Report

Shijie Jiang and colleagues described isolation of an efficient oil degrading strain WCO-9 from an oil-contaminated soil using olive oil as the sole carbon source. The strain was identified as Acinetobacter junii WCO-9 by using 16SrDNA sequence and NCBI blast analysis. The enzyme production of this strain was investigated by testing the activity of lipase towardρ-nitrophenyl decanoate, and the degradatioin capability of WCO-9 toward a variety of edible oils was investigated via measure the size of transparent circle. Finally, a large amount of work around genome comparison and pan-genome analysis of WCO-9, and lipase related genes and triacylglycerol metabolic pathways were also analyzed. The paper is well organized and includes some interesting data and results. However, the results described in this paper are hard to confirm the application potency of WCO-9 in oil dagradation, because the actual oil degradability of this strain on different oil was not tested, transparent circle method can be used for a preliminary analysis of lipase activity, but the oil degradability of WCO-9 cannot be fully proved. Then, the manuscript cannot be accepted in its current form, and a major revision is needed.

 Comments:

1. The actual oil degradability (the oil degradation rate) of this strain should be provided.

2. The title of this paper should be corrected as “Biodegradation of oil by a newly isolated strain Acinetobacter junii WCO-9 and its comparative Pan-genome analysis”, since the comparative Pan-genome analysis cannot reveal the high efficient oil degradation capability of WCO-9, just reveal the possible related oil metabolism genes in WCO-9.

Reviewer 2 Report

Microorganisms 2167671

Authors

Shape comments

In line 327, when the author and collaborators were cited, add the year

Comments or clarifications

What was the methodology for the identification and comparison o lipase genes and triacylglycerol metabolic pathways?

Bioinformatics validation of genomic parameters had replicates?

Sincerely

Reviewer 3 Report

Biological treatment modalities for environmental wastes have important applications, and microorganisms with degradation capabilities for environmental wastes have played an important role in bioremediation pathways in recent years. This study addresses the problem of waste cooking oil, and the screening of oil degrading bacteria WCO-9 from an oily environment is of potential application. The highly efficient oil degradation capability of WCO-9 was clarified by developing a comparative assay with WCO-9 and A. juniiATCC 17908 (a control strain). Comparative analysis of oil degradation pathways and lipase genes, combined with pan-genome analysis well revealed the reasons for the uniqueness of WCO-9 and the formation of efficient oil degradation ability, with clear purpose and logic.

1.       The software package should be included in the materials and methods section. Please supplement.

2.       The “sp.” of Acinetobacter sp. shouldn't be italics.

3.       The other ρ-nitrophenyl ester substrate should be included in the materials and method section. Please supplement.

4.       Abstract: conclusion should be more specific based on the findings.

5.       Page 1, line 43,44, the first letter of Such……” and Flocculation should be lowercase.

6.       Page 1, line 24, “its enzyme activity of ρ-nitrophenyl decanoate (ρ-NPD) decomposition was 3000 U/L”. But, page 6, line 11, “the hydrolytic activity of ρ-NPD substrate produced by WCO-9 was 3040 U/mL”. Please check and revise.

7.       The conclusion does not explain the title well. Please revise the conclusions.

8.       In general, most comparative genomic analyses use different strains of the same genus, but why strains of the same species were chosen for comparison in this study, and whether the same species strains used are representative and can explain the problem?

9.       WCO-9 can degrade a variety of oils and with high lipase activity, which can be further applied to the removal of waste oils in the environment. In addition, it is suggested that the subsequent functional analysis of lipase genes, especially molecular chaperones and specific lipases, is of great significance for the industrial application of lipases.

Reviewer 4 Report

A paper reports a novel strain of Acinetobacter junii capable of oil (lipid) degradation and its genome composition, which confirmed the presence of lipase and chaperone genes absent in other non-oil-degrading Acinetobacter spp. While this topic is not new and “Acinetobacter spp. have been heavily studied as an important source of alkaline and low-temperature lipases [50-52]”, the authors obtained some new results, combining physiological features and genome mining.  The paper is scientifically sound and easy to read. All methods are described in details (some of them should be amended with relevant references) and the results are statistically justified.

Specific comments

P45 chemical methods (Flocculation, electrolysis) – these are rather physico-chemical methods.

P69-70 Acinetobacter genus is currently best reported in terms of infection and resistance to coliphages and carbapenems represented by Acinetobacter baumannii [24,25]. - Acinetobacter junii is an opportunistic pathogen that mainly affects patients who have had prior antimicrobial therapy, invasive procedures, or malignancy (Hung YT, Lee YT, Huang LJ, et al. Clinical characteristics of patients with Acinetobacter junii infection. J Microbiol Immunol Infect. 2009;42(1):47-53.). However, there is at least one report of A. junii urinary tract infection (Abo-Zed A, Yassin M, Phan T. Acinetobacter junii as a rare pathogen of urinary tract infection. Urol Case Rep. 2020;32:101209). So this species would be not recommended as a bioremediation agent.

P94 sp. should be not Italic.

P138-150 References should be provided to the methods of assessing lipase activity and oil degradation using oil plates.

P394 Waste cooking oil is much less biodegradable than native oils, moreover the presence of waste cooking oil can inhibit or reduce the biodegradation rate, so it should be tested specially.

Reviewer 5 Report

COMMENTS FOR THE AUTHORS

Manuscript Title: Comparative Pan-genome Analysis Reveals Highly Efficient  Oil Degradation Capabilities of Acinetobacter junii WCO-9.

I found the manuscript very interesting. Only few issues should be addressed before the manuscript can be published.

1. The Authors found 11 lipase genes and 3  lipase chaperone genes in the genome of Acinetobacter WCO-9. Somehow  I can see all 3 lipase chaperone genes but only 4 lipase genes (I am refering to the genome deposited in GenBank). But I also checked randomly several other genomes and usually found only 1 or 2 lipase genes. Does it mean that some lipase genes are not correctly annotated in genomes  and are marked as “hypothetical protein”. Or the Authors include also phospholipase genes?

2. If a measured value from the activity of the strain ATCC 17908 was zero (no oil degradation was observed), presenting SE or SD values makes no sense (Table 1)

3. I am not very familiar with KEGG annotation but what are possible genes in these particular bacterial genomes responsible for cancers, immune diseases, nervous system, neurodegenerative diseases and overview (Fig S6B)? Perhaps it would be better to remove this  vague  information from the chart? Especially “overview” as a functional category looks weird.

4. As for Figure 4. I don’t think strains isolated from pus or urine are from “water environment” My guess they are just opportunistic pathogens.
